# Abstract Diagrammatic Reasoning with Multiplex Graph Networks

**Duo Wang & Mateja Jamnik & Pietro Lio**
Department of Computer Science and Technology
University of Cambridge
Cambridge, United Kingdom
{Duo.Wang,Mateja.Jamnik,Pietro.Lio}@cl.cam.ac.uk

## Abstract

Abstract reasoning, particularly in the visual domain, is a complex human ability, but it remains a challenging problem for artificial neural learning systems. In this work we propose MXGNet, a multilayer graph neural network for multi-panel diagrammatic reasoning tasks. MXGNet combines three powerful concepts, namely, object-level representation, graph neural networks and multiplex graphs, for solving visual reasoning tasks. MXGNet first extracts object-level representations for each element in all panels of the diagrams, and then forms a multi-layer multiplex graph capturing multiple relations between objects across different diagram panels. MXGNet summarises the multiple graphs extracted from the diagrams of the task, and uses this summarisation to pick the most probable answer from the given candidates. We have tested MXGNet on two types of diagrammatic reasoning tasks, namely Diagram Syllogisms and Raven Progressive Matrices (RPM). For an Euler Diagram Syllogism task MXGNet achieves state-of-the-art accuracy of 99.8%. For PGM and RAVEN, two comprehensive datasets for RPM reasoning, MXGNet outperforms the state-of-the-art models by a considerable margin.

## 1 Introduction

Abstract reasoning has long been thought of as a key part of human intelligence, and a necessary component towards Artificial General Intelligence. When presented in complex scenes, humans can quickly identify elements across different scenes and infer relations between them. For example, when you are using a pile of different types of LEGO bricks to assemble a spaceship, you are actively inferring relations between each LEGO brick, such as in what ways they can fit together. This type of abstract reasoning, particularly in the visual domain, is a crucial key to human ability to build complex things.

Many tests have been proposed to measure human ability for abstract reasoning. The most popular test in the visual domain is the Raven Progressive Matrices (RPM) test (Raven (2000)). In the RPM test, the participants are asked to view a sequence of contextual diagrams, usually given as a $3 \times 3$ matrices of diagrams with the bottom-right diagram left blank. Participants should infer abstract relationships in rows or columns of the diagram, and pick from a set of candidate answers the correct one to fill in the blank. Figures 1 (a) shows an example of RPM tasks containing XOR relations across diagrams in rows. More examples can be found in Appendix C. Another widely used test for measuring reasoning in psychology is Diagram Syllogism task (Sato et al. (2015)), where participants need to infer conclusions based on 2 given premises. Figure 1c shows an example of Euler Diagram Syllogism task.

Barrett et al. (2018) recently published a large and comprehensive RPM-style dataset named Procedurally Generated Matrices 'PGM', and proposed Wild Relation Network (WReN), a state-of-the-art neural net for

---

*Corresponding Author

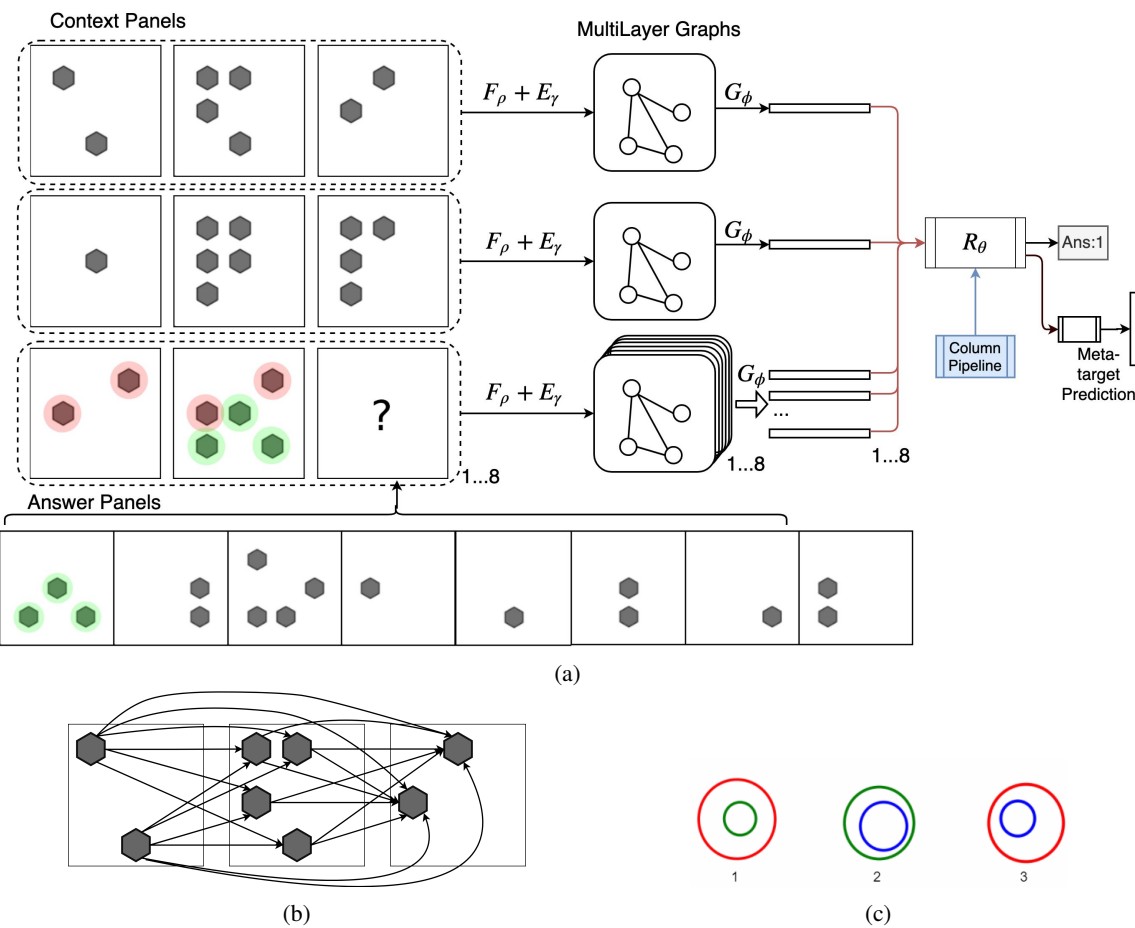

Figure 1: (a) shows an example of RPM tasks containing XOR relations across diagrams in rows and the overview of MXGNet architecture. Here $F_\rho$ is object representation module, $E_\gamma$ is edge embeddings module, $G_\phi$ is graph summarization module and $R_\theta$ is reasoning network. (b) shows an example of a multilayer graph formed from objects in the first row of diagrams in the example. (c) An example of syllogism represented in Euler diagrams.

RPM-style tasks. While WReN outperforms other state-of-the-art vision models such as Residual Network He et al. (2016), the performance is still far from deep neural nets' performance on other vision or natural language processing tasks. Recently, there has been a focus on object-level representations (Yi et al. (2018); Hu et al. (2017); Hudson & Manning (2018); Mao et al. (2019); Teney et al. (2017); Zellers et al. (2018)) for visual reasoning tasks, which enable the use of inductive-biased architectures such as symbolic programs and scene graphs to directly capture relations between objects. For RPM-style tasks, symbolic programs are less suitable as these programs are generated from given questions in the Visual-Question Answering setting. In RPM-style tasks there are no explicit questions. Encoding RPM tasks into graphs is a more natural choice. However, previous works on scene graphs (Teney et al. (2017); Zellers et al. (2018)) model a single image as graphs, which is not suitable for RPM tasks as there are many different layers of relations across different subsets of diagrams in a single task.

In this paper we introduce MXGNet, a multi-layer multiplex graph neural net architecture for abstract diagram reasoning. Here 'Multi-layer' means the graphs are built across different diagram panels, where each diagram is a layer. 'Multiplex' means that edges of the graphs encode multiple relations between different element attributes, such as colour, shape and position. Multiplex networks are discussed in detail by Kao & Porter (2018). We first tested the application of multiplex graph on a Diagram Syllogism dataset (Wang et al. (2018a)), and confirmed that multiplex graph improves performance on the original model. For RPM task, MXGNet encodes subsets of diagram panels into multi-layer multiplex graphs, and combines summarisation of several graphs to predict the correct candidate answer. With a hierarchical summarisation scheme, each graph is summarised into feature embeddings representing relationships in the subset. These relation embeddings are then combined to predict the correct answer.

For PGM dataset (Barrett et al. (2018)), MXGNet outperforms WReN, the previous state-of-the-art model, by a considerable margin. For 'neutral' split of the dataset, MXGNet achieves 89.6% test accuracy, 12.7% higher than WReN's 76.9%. For other splits MXGNet consistently performs better with smaller margins. For the RAVEN dataset (Zhang et al. (2019)), MXGNet, without any auxiliary training with additional labels, achieves 83.91% test accuracy, outperforming 59.56% accuracy by the best model with auxiliary training for the RAVEN dataset. We also show that MXGNet is robust to variations in forms of object-level representations. Both variants of MXGNet achieve higher test accuracies than existing best models for the two datasets.

## 2 RELATED WORK

**Raven Progressive Matrices**: Hoshen & Werman (2017) proposed a neural network model on Raven-style reasoning tasks that are a subset of complete RPM problems. Their model is based on Convolutional Network, and is demonstrated to be ineffective in complete RPM tasks (Barrett et al. (2018)). Mandziuk & Zychowski also experimented with an auto-encoder based neural net on simple single-shape RPM tasks. Barrett et al. (2018) built PGM, a complete RPM dataset, and proposed WReN, a neural network architecture based on Relation Network (Santoro et al. (2017)).Steenbrugge et al. (2018) replace CNN part of WReN with a pre-trained Variational Auto Encoder and slightly improved performance. Zhang et al. (2019) built RAVEN, a RPM-style dataset with structured labels of elements in the diagrams in the form of parsing trees, and proposed Dynamic Residual Trees, a simple tree neural network for learning with these additional structures. Anonymous (2020) applies Multi-head attention (Vaswani et al. (2017)), originally developed for Language model, on RPM tasks.

**Visual Reasoning**: RPM test falls in the broader category of visual reasoning. One widely explored task of visual reasoning is Visual Question Answering(VQA). Johnson et al. (2017) built CLEVR dataset, a VQA dataset that focuses on visual reasoning instead of information retrieval in traditional VQA datasets. Current leading approaches (Yi et al. (2018); Mao et al. (2019)) on CLEVR dataset generate synthetic programs using questions in the VQA setting, and use these programs to process object-level representations extracted with objection detection models (Ren et al. (2015)). This approach is not applicable to RPM-style problems as there is no explicit question present for program synthesis.

**Graph Neural Networks**: Recently there has been a surge of interest in applying Graph Neural Networks (GNN) for datasets that are inherently structured as graphs, such as social networks. Many variants of GNNs (Li et al. (2015); Hamilton et al. (2017); Kipf & Welling (2016); Veličković et al. (2017)) have been proposed, which are all based on the same principle of learning feature representations of nodes by recursively aggregating information from neighbour nodes and edges. Recent methods (Teney et al. (2017); Zellers et al. (2018)) extract graph structures from visual scenes for visual question answering. These methods build scene graphs in which nodes represent parts of the scene, and edges capture relations between these parts. Such methods are only applied to scenes of a single image. For multi-image tasks such as video classification, Wang et al. (2018b) proposed non-local neural networks, which extract dense graphs where pixels in feature maps are connected to all other feature map pixels in the space-time dimensions.

## 3 REASONING TASKS

### 3.1 DIAGRAM SYLLOGISM

Syllogism is a reasoning task where conclusion is drawn from two given assumed propositions (premises). One well-known example is 'Socrates is a man, all man will die, therefore Socrates will die'. Syllogism can be conveniently represented using many types of diagrams (Al-Fedaghi (2017)) such as Euler diagrams and Venn diagrams. Figure 1 (c) shows an example of Euler diagram syllogism. Wang et al. (2018a) developed Euler-Net, a neural net architecture that tackles Euler diagram syllogism tasks. However Euler-Net is just a simple Siamese Conv-Net, which does not guarantee scalability to more entities in diagrams. We show that the addition of multiplex graph both improves performance and scalability to more entities.

### 3.2 RAVEN PROGRESSIVE MATRICES

In this section we briefly describe Raven Progressive Matrices (RPM) in the context of the PGM dataset (Barrett et al. (2018)) and the RAVEN dataset (Zhang et al. (2019)). RPM tasks usually have 8 context diagrams and 8 answer candidates. The context diagrams are laid out in a $3 \times 3$ matrix $\mathbf{C}$ where $c_{1,1}, ..c_{3,2}$ are context diagrams and $c_{3,3}$ is a blank diagram to be filled with 1 of the 8 answer candidates $\mathbf{A} = \{a_1, \ldots, a_8\}$. One or more relations are present in rows or/and columns of the matrix. For example, in Figure 1 (a), there is $XOR$ relation of positions of objects in rows of diagrams. With the correct answer filled in, the third row and column must satisfy all relations present in the first 2 rows and columns (in the RAVEN dataset, relations are only present in rows). In addition to labels of correct candidate choice, both datasets also provide labels of meta-targets for auxiliary training. The meta-target of a task is a multi-hot vector encoding tuples of $(r, o, a)$ where $r$ is the type of a relation present, $o$ is the object type and $a$ is the attribute. For example, the meta-target for Figure 1 (a) encodes $(XOR, Shape, Position)$. The RAVEN dataset also provides additional structured labels of relations in the diagram. However, we found that structured labels do not improve results, and therefore did not use them in our implementation.

## 4 METHOD

MXGNet is comprised of three main components: an object-level representation module, a graph processing module and a reasoning module. Figure 1a shows an overview of the MXGNet architecture. The object-level representation module $F_\rho$, as the name suggests, extracts representations of objects in the diagrams as nodes in a graph. For each diagram $d_i \subset \mathbf{C} \cup \mathbf{A}$, a set of nodes $v_{i,j}; i = 1 \ldots L, j = 1 \ldots N$ is extracted where $L$ is the number of layers and $N$ is the number of nodes per layer. We experimented with both fixed and dynamically learnt $N$ values. We also experimented with an additional 'background' encoder that encodes background lines (See Appendix C for an example containing background lines) into a single vector, which can be considered as a single node. The multiplex graph module $G_\phi$, for a subset of diagrams, learns the multiplex edges capturing multiple parallel relations between nodes in a multi-layer graph where each layer corresponds to one diagram in the subset, as illustrated in Figure 1 (c). In MXGNet, we consider a subset of cardinality 3 for $3 \times 3$ diagram matrices. While prior knowledge of RPM rules allows us to naturally treat rows and columns in RPM as subsets, this prior does not generalise to other types of visual reasoning problems. Considering all possible diagram combinations as subsets is computationally expensive. To tackle this, we developed a relatively quick pre-training method to greatly reduce the search space of subsets, as described below.

**Search Space Reduction**: We can consider each diagram as node $v_i^d$ in a graph, where relations between adjacent diagrams are embedded as edges $e_{ij}^d$. Note here we are considering the graph of 'diagrams', which is different from the graph of 'objects' in the graph processing modules. Each subset of 3 diagrams in this case can be considered as subset of 2 edges. We here make weak assumptions that edges exist between adjacent diagrams (including vertical, horizontal and diagonal direction) and edges in the same subset must be adjacent (defined as two edges linking the same node), which are often used in other visual reasoning

problems. We denote the subset of edges as $\{e_{ij}^d, e_{jk}^d\}$. We use 3 neural nets to embed nodes, edges and subsets. We use CNNs to embed diagram nodes into feature vectors, and MLPs to embed edges based on node embeddings and subsets based on edge embeddings. While it is possible to include graph architectures for better accuracy, we found that simple combinations of CNNs and MLPs train faster while still achieving the search space reduction results. This architecture first embeds nodes, then embeds edges based on node embedding, and finally embed subsets based on edge embedding. The subset embeddings are summed and passed through a reasoning network to predict answer probability, similar to WReN (Barrett et al. (2018)). For the exact configuration of the architecture used please refer to Appendix A. For each subset $\{e_{ij}^d, e_{jk}^d\}$, we define a gating variable $G_{ijk}$, controlling how much does each subset contributes to the final result. In practice we use $tanh$ function, which allows a subset to contribute both positively and negatively to the final summed embeddings. In training we put L1 regularization constraint on the gating variables to suppress $G_{ijk}$ of non-contributing subsets close to zero. This architecture can quickly discover rows and columns as contributing subsets while leaving gating variables of other subsets not activated. We describe the experiment results in section 5.1. While this method is developed for discovering reasoning rules for RPM task, it can be readily applied to any other multi-frame reasoning task for search space reduction. In the rest of the paper, we hard-gate subsets by rounding the gating variables, thereby reducing subset space to only treat rows and columns as valid subsets.

We treat the first 2 rows and columns as contextual subsets $c_{i,j}$ where $i$ and $j$ are row and column indices. For the last row and column, where the answers should be filled in, we fill in each of the 8 answer candidates, and make 8 row subsets $a_i, i \subset [1, 8]$ and 8 column subsets $a_i, i \subset [1, 8]$.

The graph module then summarises the graph of objects in a subset into embeddings representing relations present in the subset. The reasoning module $R_\theta$ takes embeddings from context rows/columns and last rows/columns with different candidate answers filled in, and produce normalised probability of each answer being true. It also predicts meta-target for auxiliary training using context rows/columns. Next, we describe each module in detail.

### 4.1 Object-Level Representation

In the PGM dataset there are two types of objects, namely 'shapes' and background 'lines'. While it is a natural choice to use object-level representation on shapes as they are varying in many attributes such as position and size, it is less efficient on background lines as they only vary in colour intensity. In this section we first describe object-level representation applied to 'shapes' objects, and then discuss object-level representation on 'lines' and an alternative background encoder which performs better.

In MXGNet we experiment with two types of object-level representations for 'shapes', namely CNN grid features and representation obtained with spatial attention. For CNN grid features, we use each spatial location in the final CNN feature map as the object feature vector. Thus for each feature maps of width $W$ and height $H$, $N = W \times H$ object representations are extracted. This type of representation is used widely, such as in Relation Network (Santoro et al. (2017)) and VQ-VAE (van den Oord et al. (2017)). For representation obtained with attention, we use spatial attention to attend to locations of objects, and extract representations for each object attended. This is similar to objection detection models such as faster R-CNN (Ren et al. (2015)), which use a Region Proposal Network to propose bounding boxes of objects in the input image. For each attended location a presence variable $z_{pres}$ is predicted by attention module indicating whether an object exists in the location. Thus the total number of objects $N$ can vary depending on the sum of $z_{pres}$ variables. As object-level representation is not the main innovation of this paper, we leave exact details for Appendix A.1.

For background 'lines' objects, which are not varying in position and size, spatial attention is not needed. We experimented with a recurrent encoder with Long-Short Term Memory (Hochreiter & Schmidhuber (1997)) on the output feature map of CNN, outputting $M$ number of feature vectors. However, in the experiment

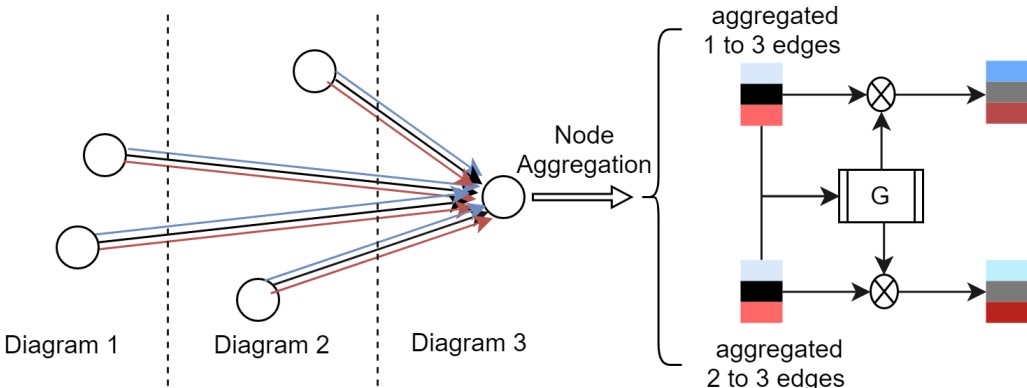

Figure 2: Illustration of multiplex edge embeddings and cross-gating function. Each edge contains a set of different sub-connections (colored differently). Multiplex edges connecting to each node in the last layer are aggregated according to its originating layer. Aggregated embeddings are then passed to a gating function $G$, which outputs gating variables from each aggregated embeddings.

we found that this performs less well than just feature map embeddings produced by feed-forward conv-net encoder.

## 4.2 MULTIPLEX GRAPH NETWORK

**Multiplex Edge Embedding**:The object-level representation module outputs a set of representations $v_{i,j}; i \subset [1, L], j \subset [1, N]$ for 'shapes' objects, where $L$ is the number of layers (cardinality of subset of diagrams) and $N$ is the number of nodes per layer. MXGNet uses an multiplex edge-embedding network $E_\gamma$ to generate edge embeddings encoding multiple parallel relation embeddings:

$$e^t_{(i,j),(l,k)} = E^t_\gamma(P^k(v_{i,j}, v_{l,k})); i \neq l, t = 1 \ldots T \tag{1}$$

Here $P^t$ is a projection layer projecting concatenated node embeddings to $T$ different embeddings. $E^t$ is a small neural net processing $t^{th}$ projections to produce the $t^{th}$ sub-layer of edge embeddings. Here, we restricted the edges to be inter-layer only, as we found using intra-layer edges does not improve performance but increases computational costs. Figure 2 illustrates these multiplex edge embeddings between nodes of different layers. We hypothesise that different layers of the edge embeddings encode similarities/differences in different feature spaces. Such embeddings of similarities/differences are useful in comparing nodes for subsequent reasoning tasks. For example,for $Progessive$ relation of object sizes, part of embeddings encoding size differences can be utilized to check if nodes in later layers are larger in size. This is similar to Mixture of Experts layers (Eigen et al. (2013); Shazeer et al. (2017)) introduced in Neural Machine Translation tasks. However, in this work we developed a new cross-multiplexing gating function at the node message aggregation stage, which is described below.

**Graph Summarisation**: After edge embeddings are generated, the graph module then summarises the graph into a feature embedding representing relations present in the subset of diagrams. We aggregate information in the graph to nodes of the last layer corresponding to the third diagram in a row or column, because in RPM tasks the relations are in the form $Diagram3 = Function(Diagram1, Diagram2)$. All edges connecting nodes in a particular layer $v_{i,j}; i \neq L$, to a node $v_{L,k}$ in the last layer $L$ are aggregated by a function $F_{ag}$ composed of four different types of set operations, namely $max, min, sum$ and $mean$:

$$fv_{i,k} = F_{ag}(e_{(i,1),(L,k)} \ldots e_{(i,1),(L,k)}); F_{ag} = concat(max(), min(), sum(), mean()) \tag{2}$$

We use multiple aggregation functions together because different sub-tasks in reasoning may require different types of summarization. For example, counting number of objects is better suited for $sum$ while checking if there is a object with the same size is better suited for $max$. The aggregated node information from each layer is then combined with a cross-multiplexing gating function. It is named 'cross-multiplexing' because each embeddings in the set are 'multiplexing' other embeddings in the set with gating variables that regulate which stream of information pass through. This gating function accepts a set of summarised node embeddings $\{fv_{1,k} \ldots fv_{N,k}\}$ as input, and output gating variables for each layer of node embeddings in the set:

$$\mathbf{g}_{1,k} \ldots \mathbf{g}_{N,k} = G(fv_{1,k} \ldots fv_{N,k}); \mathbf{g}_{i,k} = \{g_{i,k}^1 \ldots g_{i,k}^T\} \tag{3}$$

In practice $G$ is implemented as an MLP with multi-head outputs for different embeddings, and Sigmoid activation which constrains gating variable $g$ within the range of 0 to 1. The node embeddings of different layers are then multiplied with the gating variables, concatenated and passed through a small MLP to produce the final node embeddings: $fv_k = MLP(concat(\{fv_{i,k} \times g_{(i,k)} | i = 1 \ldots N\}))$. Node embeddings and background embeddings are then concatenated and processed by a residual neural block to produce final relation feature embeddings $r$ of the diagram subset.

### 4.3 REASONING NETWORK

The reasoning network takes relation feature embeddings $r$ from all graphs, and infers the correct answer based on these relation embeddings. We denote the relation embeddings for context rows as $r_i^{cr}; i = 1, 2$ and context columns as $r_i^{cc}; i = 1, 2$. The last row and column filled with each answer candidate $a_i$ are denoted $r_i^{ar}; i = 1, \ldots, 8$ and $r_i^{ac}; i = 1, \ldots, 8$. For the RAVEN dataset, only row relation embeddings $r^{cr}$ and $r^{ar}$ are used, as discussed in Section 3.2. The reasoning network $R_\theta$ is a multi-layer residual neural net with a softmax output activation that processes concatenated relation embeddings and outputs class probabilities for each answer candidate. The exact configuration of the reasoning network can be found in Appendix A.3.

For meta-target prediction, all relation information is contained in the context rows and columns of the RPM task. Therefore, we apply a meta-predicting network $R_{meta}$ with Sigmoid output activation to all context rows and columns to obtain probabilities of each meta-target categories:

$$p_{meta} = R_{meta}(r_1^{cr} + r_2^{cr} + r_1^{cc} + r_2^{cc}) \tag{4}$$

### 4.4 TRAINING

The full pipeline of MXGNet is end-to-end trainable with any gradient descent optimiser. In practice, we used RAdam optimiser (Liu et al. (2019)) for its fast convergence and robustness to learning rate differences. The loss function for the PGM dataset is the same as used in WReN (Barrett et al. (2018)): $\mathcal{L} = \mathcal{L}_{ans} + \beta\mathcal{L}_{meta-target}$ where $\beta$ balances the training between answer prediction and meta-target prediction. For the RAVEN dataset, while the loss function can include auxiliary meta-target and structured labels as $\mathcal{L} = \mathcal{L}_{ans} + \alpha\mathcal{L}_{struct} + \beta\mathcal{L}_{meta-target}$, we found that both auxiliary targets do not improve performance, and thus set $\alpha$ and $\beta$ to 0.

## 5 EXPERIMENTS

### 5.1 SEARCH SPACE REDUCTION

The Search Space Reduction model is applied on both PGM and RAVEN dataset to reduce the subset space. After 10 epochs, only gating variables of rows and columns subset for PGM and of rows for RAVEN have value larger than 0.5. The Gating variables for three rows are 0.884, 0.812 and 0.832. The gating variables for three columns are 0.901, 0.845 and 0.854. All other gating variables are below the threshold value of 0.5. Interestingly all activated (absolute value $> 0.5$) gating variables are positive. This is possibly because it is easier for the neural net to learn an aggregation function than a comparator function. Exact experiment statistics can be found in Appendix D.

## 5.2 DIAGRAM SYLLOGISM PERFORMANCE

We first test how well can the multiplex graph network capture relations for the simple Diagram Syllogism task. We simply add the multiplex graph to the original Conv-Net used in (Wang et al. (2018a)). MXGNet achieved 99.8% accuracy on both 2-contour and 3-contour tasks, higher than the original paper's 99.5% and 99.4% accuracies. The same performance on 2-contour and 3-contour tasks also show that MXGNet scales better for more entities in the diagram. For more details please refer to Appendix E.

## 5.3 RPM TASK PERFORMANCES

In this section we compare all variants of MXGNet against the state-of-the-art models for the PGM and the RAVEN datasets. For the PGM dataset, we tested against results of WReN (Barrett et al. (2018)) in the auxiliary training setting with $\beta$ value of 10. In addition, we also compared MXGNet with VAE-WReN (Steenbrugge et al. (2018))'s result without auxiliary training. For the RAVEN dataset, we compared with WReN and ResNet model's performance as reported in the original paper (Zhang et al. (2019)). We evaluated MXGNet with different object-level representations (Section 4.1) on the test data in the 'neutral' split of the PGM dataset.

Table 1 (a) shows test accuracies of model variants compared with WReN and VAE-WReN for the case without auxiliary training ($\beta = 0$) and with auxiliary training ($\beta = 10$) for the PGM dataset. Both model variants of MXGNet outperform other models by a considerable margin, showing that the multi-layer graph is indeed a more suitable way to capture relations in the reasoning task. Model variants using grid features from the CNN feature maps slightly outperform model using spatial-attention-based object representations for both with and without auxiliary training settings. This is possibly because the increased number of parameters for the spatial attention variant leads to over-fitting, as the training losses of both model variants are very close. In our following experiments for PGM we will use model variants using CNN features to report performances.

Table 1 (b) shows test accuracies of model variants compared with WReN the best performing ResNet models for RAVEN dataset. WReN surprisingly only achieves 14.69% as tested by Zhang et al. (2019). We include results of the ResNet model with or without Dynamic Residual Trees (DRT) which utilise additional structure labels of relations. We found that for the RAVEN dataset, auxiliary training of MXGNet with meta-target or structure labels does not improve performance. Therefore, we report test accuracies of models trained only with the target-prediction objective. Both variants of MXGNet significantly outperform the ResNet models. Models with spatial attention object-level representations under-perform simpler CNN features slightly, most probably due to overfitting, as the observed training losses of spatial attention models are in fact lower than CNN feature models.

## 5.4 GENERALISATION EVALUATION FOR PGM

In the PGM dataset, other than the neutral data regime in which test dataset's sampling space is the same as the training dataset, there are also other data regimes which restrict the sampling space of training or test data to evaluate the generalisation capability of a neural network. In the main paper, due to space limitations, we selected 2 representative regimes, the 'interpolation' regime and the 'extrapolation' regime to report results. For results of other data splits of PGM, please refer to Appendix G. For 'interpolation' regime, in the training dataset, when attribute $a = color$ and $a = size$, the values of $a$ are restricted to even-indexed values in the spectrum of $a$ values. This tests how well can a model 'interpolate' for missing values. For 'Extrapolation' regime, in the training dataset, the value of $a$ is restricted to be the lower half of the value spectrum. This tests how well can a model 'extrapolate' outside of the value range in the training dataset. Table 2 shows validation and test accuracies for all three data regimes with and without auxiliary training. In addition, differences between validation and test accuracies are also presented to show how well can models generalise. MXGNet models consistently perform better than WReN for all regimes tested. Interesting for 'Interpolation' regime, while validation accuracy of MXGNet is lower than WReN, the test accuracy is higher. In addition, for regime

| Model | WReN Barrett et al. (2018) | VAE-WReN Steenbrugge et al. (2018) | ARNe Anonymous (2020) | MXGNet CNN    Sp-Attn | |
|---|---|---|---|---|---|
| acc. (%)$\beta = 10$ | 76.9 | N/A | 88.2 | **89.6** | 88.8 |
| acc. (%)$\beta = 0$ | 62.6 | 64.2 | N/A | **66.7** | 66.1 |

(a) PGM

| Model | WReN Zhang et al. (2019) | ResNet Zhang et al. (2019) | ResNet+DRT Zhang et al. (2019) | ARNe Anonymous (2020) | MXGNet CNN    Sp-Attn | |
|---|---|---|---|---|---|---|
| acc. (%) | 14.69 | 53.43 | 59.56 | 19.67 | **83.91** | 82.61 |

(b) RAVEN

Table 1: (a) shows results comparing MXGNet model variants against WReN for the PGM dataset. (b) shows results comparing MXGNet model variants against ResNet models for the RAVEN dataset. The object-level representation has two variations which are (o1) CNN features and (o2) Spatial Attention features (Section 4.1).

'Interpolation' and 'Extrapolation', MXGNet also shows a smaller difference between validation and test accuracy. These results show that MXGNet has better capability of generalising outside of the training space.

| **Model** | **Regime** | $\beta = 0$ | | | $\beta = 10$ | | |
|---|---|---|---|---|---|---|---|
| | | **Val.(%)** | **test%** | **Diff.** | **Val.(%)** | **test%** | **Diff.** |
| WReN | Neutral | 63.0 | 62.6 | -0.4 | 77.2 | 76.9 | -0.3 |
| | Interpolation | 79.0 | 64.4 | -14.6 | 92.3 | 67.4 | -24.9 |
| | Extrapolation | 69.3 | 17.2 | -52.1 | 93.6 | 15.5 | -79.1 |
| MXGNet | Neutral | 67.1 | **66.7** | -0.4 | 89.9 | **89.6** | -0.3 |
| | Interpolation | 74.2 | **65.4** | -8.8 | 91.5 | **84.6** | -6.9 |
| | Extrapolation | 69.1 | **18.9** | -50.2 | 94.3 | **18.4** | -75.9 |

Table 2: Generalisation performance comparing MXGNet model variants against WReN. **'Diff.'** is the difference between the test and the validation performances.

## 6 DISCUSSION AND CONCLUSION

We presented MXGNet, a new graph-based approach to diagrammatic reasoning problems in the style of Raven Progressive Matrices (RPM). MXGNet combines three powerful ideas, namely, object-level representation, graph neural networks and multiplex graphs, to capture relations present in the reasoning task. Through experiments we showed that MXGNet performs better than previous models on two RPM datasets. We also showed that MXGNet has better generalisation performance.

One important direction for future work is to make MXGNet interpretable, and thereby extract logic rules from MXGNet. Currently, the learnt representations in MXGNet are still entangled, providing little in the way of understanding its mechanism of reasoning. Rule extraction can provide people with better understanding of the reasoning problem, and may allow neural networks to work seamlessly with more programmable traditional logic engines.

While the multi-layer multiplex graph neural network is designed for RPM style reasoning task, it can be readily extended to other diagrammatic reasoning tasks where relations are present between multiple elements

across different diagrams. One example of a real-world application scenario is robots assembling parts of an object into a whole, such as building a LEGO model from a room of LEGO blocks. MXGNet provides a suitable way of capturing relations between parts, such as ways of piecing and locking two parts together.

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

# A  ARCHITECTURE

In this section we present exact configurations of all model variants of MXGNet. Due to the complexity of architectures, we will describe each modules in sequence. The object-level representation has two variations which are (o1) CNN features and (o2) Spatial Attention features. Also the models for PGM and RAVEN dataset differ in details. Unless otherwise stated, in all layers we apply Batch Normalization Ioffe & Szegedy (2015) and use Rectified Linear Unit as activation function.

## A.1  OBJECT-LEVEL REPRESENTATION ARCHITECTURE

**CNN features**: The first approach applies a CNN on the input image and use each spatial location in the final CNN feature map as the object feature vector. This type of representation is used widely, such as in Relation Network Santoro et al. (2017) and VQ-VAE van den Oord et al. (2017). Formally, the output of a CNN is a feature map tensor of dimension $H \times W \times D$ where $H$, $W$ and $D$ are respectively height, width and depth of the feature map. At each $H$ and $W$ location, an object vector is extracted. This type of object representation is simple and fast, but does not guarantee that the receptive field at each feature map location fully bounds objects in the image.

We use a residual module He et al. (2016) with two residual blocks to extract CNN features, as shown in figure 4.This is because Residual connections show better performance in experiments. The structure of a single Residual Convolution Block is shown in figure 3.Unless otherwise stated, convolutional layer in residual blocks has kernel size of $3 \times 3$. The output feature map processed by another residual block is treated as background encoding because we found that convolutional background encoding gives better results than feature vectors.

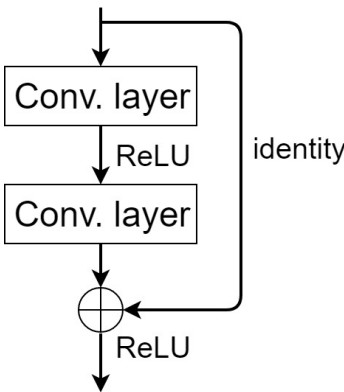

Figure 3: Architecture of a single Residual Convolution Block.

**Spatial Attention Object-level representation**: The second approach is to use spatial attention to attend to locations of objects, and extract representations for each object attended. This is similar to object detection models such as faster R-CNN Ren et al. (2015), which use a Region Proposal Network to propose bounding boxes of objects in the input image. In practice, we use Spatial Transformer Jaderberg et al. (2015) as our spatial attention module. Figure 5 shows the architecture used for extracting object-level representation using spatial attention. A CNN composed of 1 conv layr and 2 residual blocks is first applied to the input image, and the last layer feature map is extracted. This part is the same as CNN grid feature module. A spatial attention network composed of 2 conv layer then processes information at each spatial location on the feature map, and outputs $k$ numbers of $z = (z^{pres}, z^{where})$, corresponding to $k$ possible objects at each location. Here, $z^{pres}$ is a binary value indicating if an object exists in this location, and $z^{where}$ is an affine

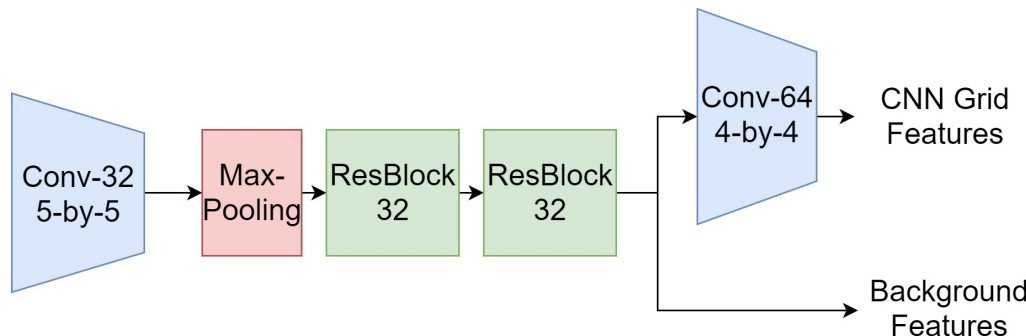

Figure 4: CNN feature object-level representation module. 'Conv' is convolution layers, 'Max-Pooling' is max-pooling layer and 'ResConv Block' is Residual Convolutional Block.

transformation matrix specifying a sampling region on the feature maps. $z^{pres}$, the binary variable, is sampled from Gumbel-Sigmoid distribution Maddison et al. (2016); Jang et al. (2016), which approximates the Bernoulli distribution. We set Gumbel temperature to 0.7 throughout the experiments. For the PGM dataset we restricted $k$ to be 1 and $z^{where}$ to be a translation and scaling matrix as 'shapes' objects do not overlap and do not have affine transformation attributes other than scaling and translation. For all $z_i; i \subset [1, H \times W]$, if $z_i^{pres}$ is 1, an object encoder network samples a patch from location specified by $z_i^{where}$ using a grid sampler with a fixed window size of $4 \times 4$ pixels. More details of the grid sampler can be found in Jaderberg et al. (2015). The sampled patches are then processed by a conv-layer to generate object embeddings.

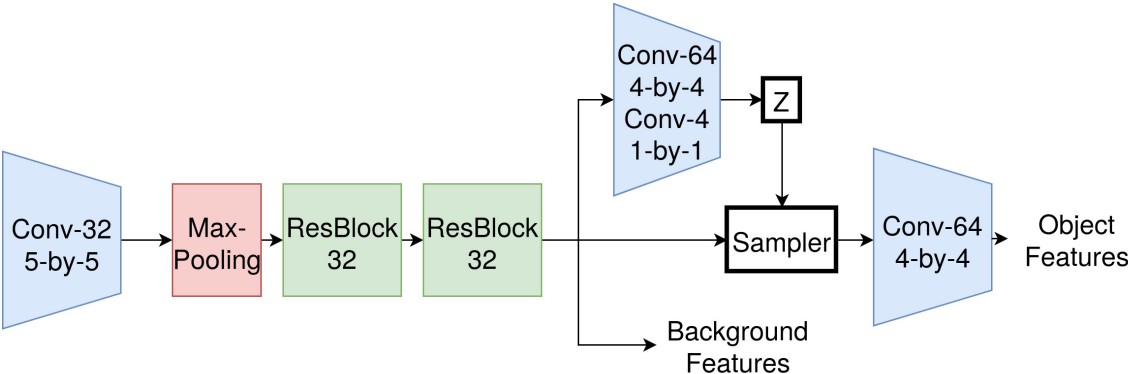

Figure 5: Spatial attention based feature object-level representation module. 'Conv' is convolution layers, 'Max-Pooling' is max-pooling layer and 'ResConv Block' is Residual Convolutional Block. $z$ is the spatial attention variable $(z^{pres}, z^{where})$. Sampler is a grid sampler which samples grid of points from given feature maps.

## A.2 GRAPH NETWORKS

**Multiplex Edge Embeddings**:Figure 2 in the main paper shows an overview of the multiplex graph architecture. While motivation and overview of architecture is explained in section 4.2 of the main paper, in this section we provide exact configurations for each part of the model. Each sub-layer of the multiplex edge is

embedded by a small MLP. For PGM dataset, we use 6 parallel layers for each multiplex edge embeddings , with each layer having 32 hidden units and 8 output units. For RAVEN dataset we use 4 layers with 16 hidden units and 8 output units because RAVEN dataset contains fewer relations types than PGM dataset. Gating function is implemented as one Sigmoid fully connected layer with hidden size equal to the length of concatenated aggregated embeddings. Gating variables are element-wise multiplied with concatenated embeddings for gating effects. Gated embeddings are then processed with a final fully connected layer with hidden size 64.

**Graph Summarization**: This module summarizes all node summary embeddings and background embeddings to produce a diagram subset embedding representing relations present in the set of diagrams. We experimented with various approaches and found that keeping embeddings as feature maps and processing them with residual blocks yields the best results. Background feature map embeddings are generated with one additional residual block of $48$ on top of lower layer feature-extracting resnet. For object representations obtained from CNN-grid features, we can simply reshape node embeddings into a feature map, and process it with additional conv-nets to generate a feature map embeddings of the same dimension to background feature map embeddings. For object representations with spatial attention, we can use another Spatial Transformer to write node summary embeddings to its corresponding locations on a canvas feature map. Finally we concatenate node summary embeddings and background embeddings and process it with 2 residual blocks of size $64$ to produce the relation embeddings.

### A.3 REASONING NETWORK

Figure 6 shows the reasoning network configuration for RPM tasks. We experimented with the approach introduced in Barrett et al. (2018), which compute scores for each answer candidates and finally normalize the scores. We found this approach leads to severe overfitting on the RAVEN dataset, and therefore used a simpler approach to just concatenate all relation embeddings and process them with a neural net. In practice we used two residual blocks of size 128 and 256, and a final fully connected layer with 8 units corresponding to 8 answer candidates. The output is normalized with softmax layer. For Meta-target prediction, all context relation embeddings (context rows and columns for PGM while only rows for RAVEN dataset) are summed and fed into a fully connected prediction layer with Sigmoid activation. For PGM there are 12 different meta-targets while for RAVEN there are 9.

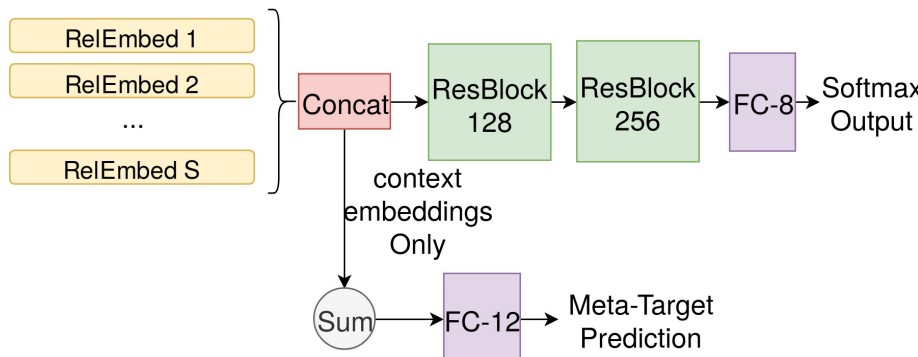

Figure 6: Architecture overview of reasoning module. 'RelEmbed' is relation embeddings, 'Concat' is concatenation layer. 'ResBlock' is Residual Convolutional Block. 'FC' is fully connected layer.

## B  Training Details

The architecture is implemented in Pytorch framework. During training, we used RAdam optimizer Liu et al. (2019) with learning rate 0.0001, $\beta_1 = 0.9, \beta_2 = 0.999$. We used batch size of 64, and distributed the training across 2 Nvidia Geforce Titan X GPUs. We early-stop training when validation accuracy stops increasing.

## C  More Details of RPM Datasets

In PGM dataset there are two types of elements present in the diagram, namely shapes and lines. These elements have different attributes such as colour and size. In the PGM dataset, five types of relations can be present in the task: $\{Progression, AND, OR, XOR, ConsistentUnion\}$. The RAVEN dataset, compared to PGM, does not have logic relations $AND, OR, XOR$, but has additional relations $Arithmetic, Constant$. In addition RAVEN dataset only allow relations to be present in rows.

Figure 7a and 7b show two examples from the PGM dataset(Image courtesy Barrett et al. (2018)). The first example contains a 'Progression' relation of the number of objects across diagrams in columns. The second examples contains a 'XOR' relation of position of objects across diagrams in rows.

In addition to shape objects, diagrams in the PGM dataset can also contain background line objects that appear at fixed locations. Figure 8a and 8b show two examples of PGM tasks containing line objects.

## D  More details on Search Space Reduction

In this section we provide detailed architecture used for Search Space reduction, and present additional experimental results.

The node embeddings are generated by applying a Conv-Net of 4 convolutional layer (32 filters in each layer) of kernel size 3, and a fully connected layer mapping flattened final-layer feature maps to a feature vector of size 256. Edge embeddings are generated by a 3-layer MLP of $512 - 512 - 256$ hidden units. Subset embeddings are generated by a fully connected layer of 512 units. The subset embeddings are gated with the gating variables and summed into a feature vector, which is then feed into the reasoning net, a 3-layer MLP with $256 - 256 - 13$. The output layer contains 13 units. The first unit gives probability of currently combined answer choice being true. The rest 12 units give meta-target prediction probabilities. This is the same as Barrett et al. (2018). The training loss function is:

$$\mathcal{L} = \mathcal{L}_{ans} + \beta \mathcal{L}_{meta-target} + \lambda \left\| \sum_{(i,j,k) \subset S} G_{i,j,k} \right\|_{L1} \tag{5}$$

In our experiment we have tested various values of $\lambda$, and found 0.01 to be the best. This model is trained with RAdam optimizer with learning rate of 0.0001 and batch size of 64. After 10 epochs of training, only gating variables of subsets that are rows and columns are above the 0.5 threshold. The Gating variables for three rows are 0.884, 0.812 and 0.832. The gating variables for three columns are 0.901, 0.845 and 0.854. All other gating variables are below 0.5. Among these, the one with highest absolute value is 0.411. Table 3 shows the top-16 ranked subsets, with each subset indexed by 2 connecting edges in the subset. Figure 9 illustrates this way of indexing the subset. For example, the first column with red inter-connecting arrows is indexed as 0-3-6. This indicates that there two edges, one connecting diagram 0 and 3, and the other connecting diagram 3-6. Similarly the subset connected by blue arrows is indexed as 1-2-5. Note that 1-2-5 and 2-1-5 is different because the 1-2-5 contains edge 1-2 and 2-5 while 2-1-5 contains edges 1-2 and 1-5.

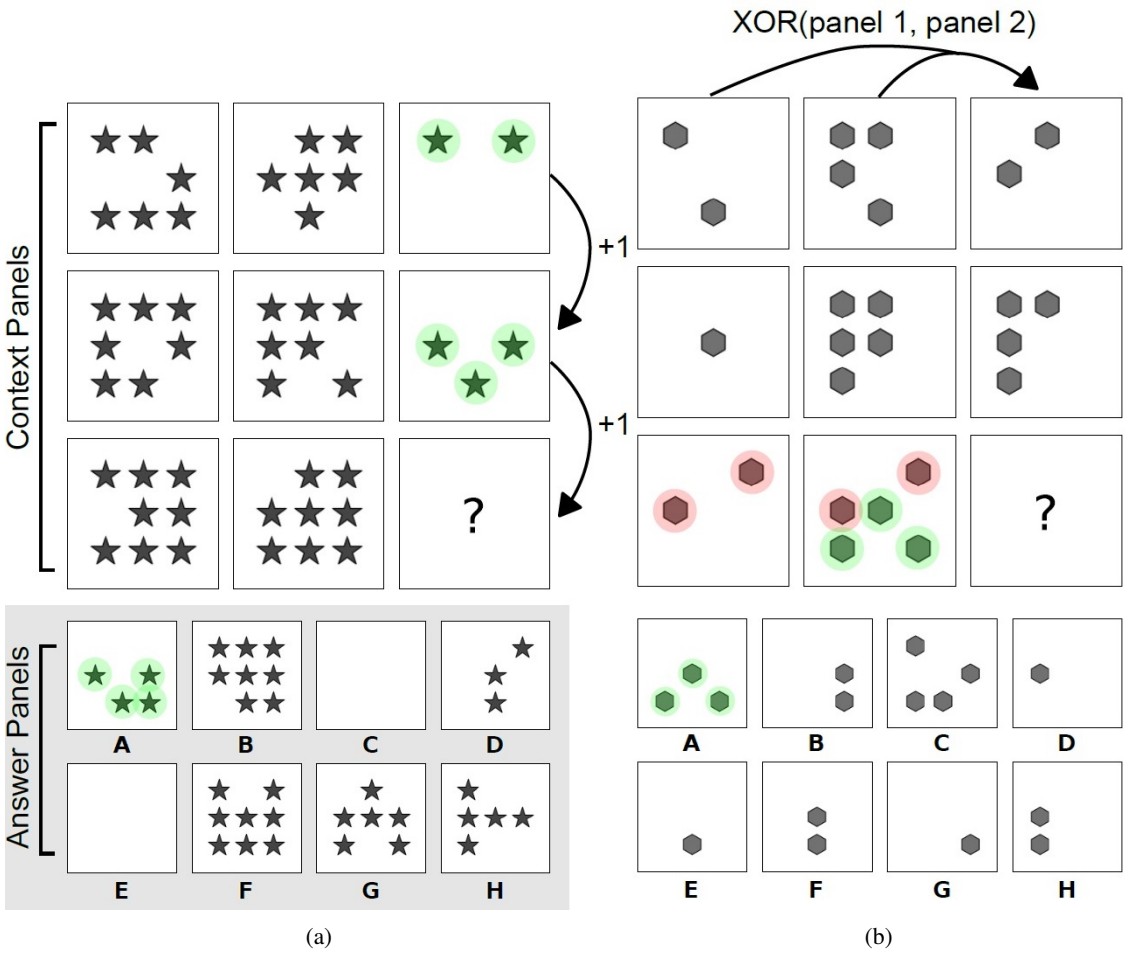

Figure 7: Two examples in PGM dataset. (a) task contains a 'Progression' relation of the number of objects across diagrams in columns while (b) contains a 'XOR' relation of position of objects across diagrams in rows.

# E    MORE DETAILS ON EULER DIAGRAM SYLLOGISM

The original model in Wang et al. (2018a) uses a Siamese Conv-Net model to process two input premise diagrams and output all consistent conclusions. Convolutional layers with shared weights are first applied to two input diagrams. The top layer feature maps are then flattened and fed into a reasoning network to make predictions. We simply use CNN grid features of the top layer feature maps as object-level representations, and use the multi-layer multiplex graph to capture object relations between the two input premise diagrams. We use a multiplex edge embeddings of 4 layers, with each layer of dimension 32. The cross-multiplexing here becomes self-multiplexing as there are only 2 diagrams (Only 1 embedding of node summary for edges from first diagram to second diagram). Final node embeddings are processed by a convolutional layer to produce the final embedding, which is also fed into the reasoning network along with the conv-net embeddings.

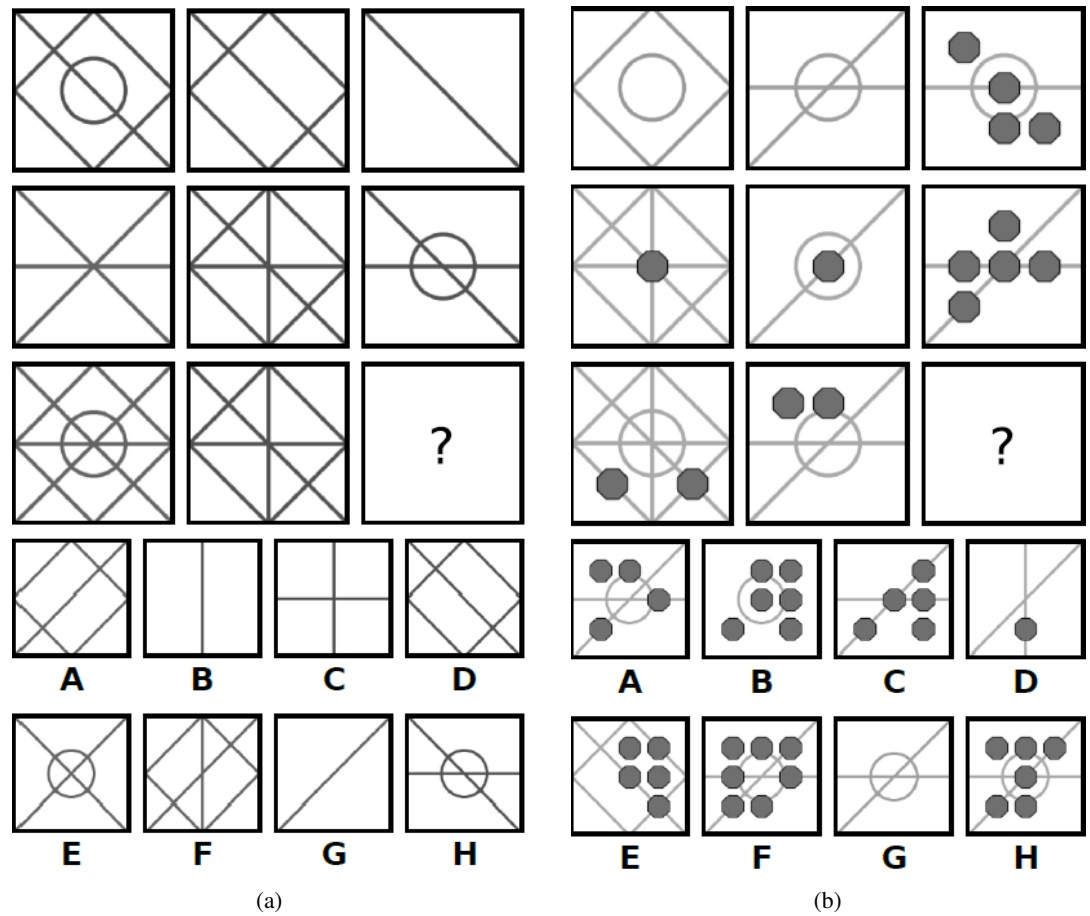

Figure 8: Two examples in PGM dataset containing background line objects.

## F   ABLATION STUDY

We performed ablation study experiments to test how much does the multiplex edges affects performance. We have tested two model variants, one without any graph modules, and the other model graphs using vanilla edge embeddings produced by MLPs, on PGM dataset. We found that without graph modules, the model only achieved 83.2% test accuracy. While this is lower than MXGNet's 89.6%, it is still higher than WReN's 76.9%. This is possibly because the search space reduction, by trimming away non-contributing subsets, allow the model to learn more efficiently. The graph model with vanilla edge embeddings achieves 88.3% accuracy, only slightly lower than MXGNet with multiplex edge embeddings. This shows that while general graph neural network is a suitable model for capturing relations between objects, the multiplex edge embedding does so more efficiently by allowing parallel relation multiplexing.

## G   ADDITIONAL GENERALIZATION PERFORMANCE ON PGM DATASET

Table 4 shows performance of MXGNet on other splits of PGM dataset. MXGNet consistently outperforms WReN for test accuracy, except for H.O. Triple Pairs and H.O. shape-color in the case $\beta = 0$ Additionally

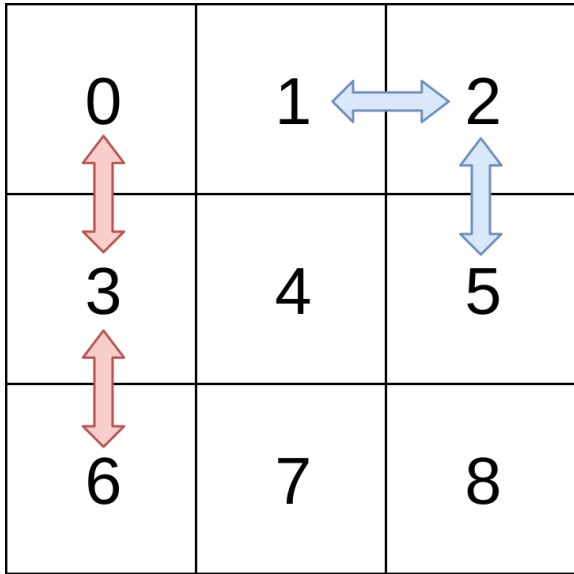

Figure 9: Illustration of diagram ordering in the matrix and numbered representation of subsets.

| Rank | Diagram subsets | $|GatingVariable|$ |
|------|-----------------|---------------------|
| 1 | 0-3-6 | 0.901 |
| 2 | 0-1-2 | 0.884 |
| 3 | 2-5-8 | 0.854 |
| 4 | 1-4-7 | 0.845 |
| 5 | 6-7-8 | 0.832 |
| 6 | 3-4-5 | 0.812 |
| 7 | 1-2-5 | 0.411 |
| 8 | 2-1-5 | 0.384 |
| 9 | 3-6-7 | 0.381 |
| 10 | 3-7-4 | 0.364 |
| 11 | 6-3-7 | 0.360 |
| 12 | 1-5-4 | 0.357 |
| 13 | 0-4-6 | 0.285 |
| 14 | 3-4-7 | 0.282 |
| 15 | 1-3-4 | 0.273 |
| 16 | 1-4-5 | 0.271 |

Table 3: All subsets ranked by the absolute value of their corresponding gating variables.

here we provide the analysis according to Sec 4.2 and Sec 4.6 in Barrett et al. (2018). unfortunately sec 4.3 of this paper, namely the analysis of distractors, cannot be performed as the publicly available dataset does not include any ground truth labels about distractors, nor any labels of present objects that can be used to synthesize distractor labels. For Meta-target prediction, MXG-Net achieves 84.1% accuracy. When Meta-target is correctly predicted, the model's target prediction accuracy increases to 92.4%. When Meta-target is incorrectly predicted, the model only has 75.6% accuracy. For three logical relations the model performs best for $OR$ relation (95.3%), and worst for $XOR$ relation(92.6%). Accuracy for line-type tasks (86.5%) is only

slightly better than for shape tasks (80.1%), showing that object representation with graph modeling does improve on relations between shapes. The type of relation with worst performance is $ConsistentUnion$, with only 75.1% accuracy. This is expected as $ConsistentUnion$ is in fact a memory task instead of relational reasoning task.

| Model | Regime | $\beta = 0$ | | | $\beta = 10$ | | |
|-------|--------|-------------|--------|-------|--------------|--------|-------|
| | | **Val.(%)** | **test%** | **Diff.** | **Val.(%)** | **test%** | **Diff.** |
| WReN | H.O. Attribute Pairs | 46.7 | 27.2 | -19.5 | 73.4 | 51.7 | -21.7 |
| | H.O. Triple Pairs | 63.9 | 41.9 | -22.0 | 74.5 | 56.3 | -18.2 |
| | H.O. Triples | 63.4 | 19.0 | -44.4 | 80.0 | 20.1 | -59.9 |
| | H.O. `line-type` | 59.5 | 14.4 | -45.1 | 78.1 | 16.4 | -61.7 |
| | H.O. `shape-color` | 69.3 | 17.2 | -52.1 | 93.6 | 15.5 | -78.1 |
| MXGNet | H.O. Attribute Pairs | 68.3 | 33.6 | -34.7 | 81.9 | 69.3 | -12.6 |
| | H.O. Triple Pairs | 67.1 | 43.3 | -23.8 | 78.1 | 64.2 | -13.9 |
| | H.O. Triples | 63.7 | 19.9 | -43.8 | 80.5 | 20.2 | -60.3 |
| | H.O. `line-type` | 60.1 | 16.7 | -43.4 | 85.2 | 16.8 | -61.5 |
| | H.O. `shape-color` | 68.5 | 16.6 | -51.9 | 89.2 | 15.6 | -73.6 |

Table 4: Generalisation performance comparing MXGNet model variants against WReN. **'Diff.'** is the difference between the test and the validation performances.

