# OpenReview forum: "Abstract Diagrammatic Reasoning with Multiplex Graph Networks"
_ICLR.cc/2020/Conference — Accept (Poster)_

### Official Review · AnonReviewer3 · 2019-10-23
**Official Blind Review #3**

**Rating:** 6

**Review:**

This paper proposes using a new version of graph networks – multiplex graph networks – which do object representation followed by some form of graph processing and reasoning to answer "IQ test" style diagrammatic reasoning, in particular including Raven Progressive Matrices that have been previously studied (a little).

The paper shows very strong results on multiple datasets, much stronger than previous results (from strong groups) on these datasets. On these grounds, I believe the paper should be accepted.

However, the structure and writing of the paper was very frustrating to me. The paper just didn't make much of an attempt to explain and then motivate/analyze the model used. I mean, if I were writing the paper, I would have considered and done many things, such as:

 - shortening the introduction
 - shortening the related work
 - making the presentation of the datasets more succinct
 - having only one figure that covers most of what is currently in figures 1 and 2
 - putting details of what seem more ancillary details like the treatment of background lines objects in an appendix
 - remove Figure 3, which didn't convey much to me in the absence of more careful explanation of the model.

so that I could motivate, carefully explain, and evaluate the main model in the paper. But here, all these things fill the main text, and we're told that we have to read the appendices to understand the model....  And the presentation in the appendix is more a dump-all-the-facts presentation than a careful development of the design.

Nevertheless, the general direction of the architecture seems sound, and the results look very strong, and there are even some useful ablations in the appendix.


**Experience Assessment:**

I have read many papers in this area.

**Review Assessment: Checking Correctness Of Derivations And Theory:**

I assessed the sensibility of the derivations and theory.

**Review Assessment: Checking Correctness Of Experiments:**

I assessed the sensibility of the experiments.

**Review Assessment: Thoroughness In Paper Reading:**

I made a quick assessment of this paper.

---

> ### Author Response · Authors · 2019-11-11
> **Thank you for your valuable comments**
>
> Thank you for your valuable comments. We improved the structuring and writing quality in this revised version. We reduced non-important parts of the paper, such as dataset description, and provide more details on the architectures. We also improved the Appendix by changing some of the table architecture representation to diagrammatic representations, which should make it more readable.

---

### Official Review · AnonReviewer1 · 2019-10-27
**Official Blind Review #1**

**Rating:** 3

**Review:**

The paper proposes a novel, feedforward, end-to-end trainable, deep, neural network for abstract diagrammatic reasoning with significant improvements over the state of the art. The proposed model architecture is reasonable and is designed to exploit the information present at multiple granularities – at the level of objects in the diagram, their relations across diagrams, and diagram subsets. As a multimodule neural pipeline, it seems a reasonable design. Further, it shows significant performance gains over the state of the art.

However, the writing quality is poor and is the primary reason for my giving it a low score. The paper is difficult to read and it’s hard to figure out the terminology and it’s grounding in the problem; the high-level abstract design and design choices that address the nature of the problem from the low level details, etc.

The paper uses terminology without explaining the reason for it - for example, why is the approach called ‘Multiplex Graph Networks’? What information is being multiplexed and how? Graphs are conceptual in the proposed approach – there doesn’t seem to be any graph algorithms or graph based processing. Once the module is run for search space reduction, the set of edges or relations (node pairs) become well-defined (in adjacent rows, columns) as well diagram subsets (edge pairs). The corresponding modules are just computing vectorial embeddings. Similarly, there is no reasoning that’s taking place. Reasoning requires tokens and grammar over such tokens which is not there in this case.  The proposed model is non-interpretable.

The technical writing is loose and hand-wavy. The appendix is a lot of grammatical mistakes.

A few clarifications may be helpful:

- “The reasoning module can also be considered as another graph processing module”?

- “… we use spatial attention to iteratively attend …” – there is no iterative attention. It’s all parallel.

- What do the ‘N’ nodes in each layer correspond to? There are clearly not objects or diagram primitives as they can vary in number in each diagram.

- if interlayer connections are between objects in different layers (diagrams), what is this supposed to capture? Clearly, there may not be any unique correspondence between objects across diagrams.

- What’s a cross-multiplexing gating function? If it’s a known concept, please provide a reference else explain.

Finally, I’m open to revising my score upwards if it turns out that I’m the only one who had difficulty with the writing. The architecture design makes sense for the addressed class of problems (though the proposed network is non-interpretable and doesn’t do any reasoning nor uses graphs or graph based processing in a meaningful way), the results are good and the experimental evaluation sufficient.


**Experience Assessment:**

I do not know much about this area.

**Review Assessment: Checking Correctness Of Derivations And Theory:**

N/A

**Review Assessment: Checking Correctness Of Experiments:**

I assessed the sensibility of the experiments.

**Review Assessment: Thoroughness In Paper Reading:**

I read the paper thoroughly.

---

> ### Author Response · Authors · 2019-11-11
> **Thank you for your valuable comments (Part 1/2)**
>
> Thank you very much for your valuable comments! In the revised version, we improved the writing, removed jargon and provided better explanations for concepts. Below are explanations for a few points mentioned in your review:
>
> 1. "Why is this approach called 'Multiplex Graph Networks'? What information is being multiplexed and how?":
>
> This approach is called 'Multiplex' Graph Networks because the architecture use graph neural networks with 'Multiplex' edges, which means that edges contain multiplex sub-connections that capture relations with different attributes, such as color and size. We mentioned this in the introduction with a citation for multiplex networks(Kao & Porter 2018). 'Multiplex' here means that multiple types of relations exist in a multi-layer network. This is slightly different from the concept of 'Multiplexing' in digital electronics and communications. We improved the discussion of the naming in the revised version.
>
> 2."Once the module is run for search space reduction, the set of edges or relations (node pairs) become well-defined (in adjacent rows, columns) as well diagram subsets (edge pairs). The corresponding modules are just computing vectorial embeddings.":
>
>  The graphs exist both on a diagram level and object level. While we performed search space reduction to trim edges of the graph of 'diagrams', we still construct graphs of objects for each of the diagram subsets. This is visualized in Figure 1(b) where each object is a node, and relations are inferred between objects and embedded in the edge embeddings. Thus the corresponding modules are processing a graph of objects rather than vector embeddings. In the revised version, we improved expalanations in 'search space reduction' section to make it clearer.
>
> 3. " there is no reasoning that’s taking place. Reasoning requires tokens and grammar over such tokens which is not there in this case.  The proposed model is non-interpretable."
>
> While we agree that the model lacks certain interpretability, we argue that the model can still be considered as undertaking 'reasoning'. We followed a recent line of work (e.g., Andreas et al 2016 , Santoro et al 2017 and Perez et al 2018) which use differentiable neural modules to model relations (equivalent to grammars) between entities (equivalent to tokens). While the black-box neural modules lack interpretability, they are still performing the 'reasoning' tasks such as in Visual Question Answering and in Raven Progressive Matrices. We do agree that improving interpretability is an important direction of future work for our models.
>
> 4. "The reasoning module can also be considered as another graph processing module?":
>
> There are two hierarchical graph levels, which are graphs of diagram subsets and graph of objects in each diagram subsets. The reasoning module can be considered as processing the graphs of diagram subsets, with each diagram subset summarized by the previous graph processing module. We agree that this statement is confusing and requires substantial explanation, and thus have removed it in the revised version.
>
> 5. "... we use spatial attention to iteratively attend...":
> We use 'iteratively' here because in some work on spatial attention such as R-CNN (Ren et al 2015) and AIR models (Eslami et al 2016), there is the idea of iteratively processing each area of attention, either with an inherently iterative Recurrent Neural Nets, or a convolutional kernel that sweeps across the images. But we agree that in practice, particularly with GPUs, the attentions are run in parallel. Thus, we removed the word 'iteratively' for less confusion.
>
> 6. "What do the ‘N’ nodes in each layer correspond to? There are clearly not objects or diagram primitives as they can vary in number in each diagram."
>
> The 'N' nodes corresponds to the number of extracted object representations in each diagram. 'N' can be both static (for CNN grid features) and dynamic (for spatial attention). For CNN grid features, there are a fixed number of locations in the feature maps, and thus 'N' is fixed and equal to H*W where H and W are the width and the height of the feature maps. For spatial attention, as explained in Appendix A.1., 'N' can vary from diagram to diagram because the spatial attention module outputs variable number of object representations. This is because even though the number of attended locations is fixed, for each location a binary presence variable 'z_pres' is computed indicating if an object is present in this location. In this case, 'N' is equal to sum of all 'z_pres' variables.
>
> Additional comments addressed in next reply.

---

> > ### Author Response · Authors · 2019-11-11
> > **Thank you for your valuable comments (Part 2/2)**
> >
> > 7." If interlayer connections are between objects in different layers (diagrams), what is this supposed to capture? Clearly, there may not be any unique correspondence between objects across diagrams":
> >
> > The interlayer connections are supposed to capture relations in the attributes of objects. For example, in the simple case of 'Progression' relation of object sizes, the connections can capture the fact that objects in later layers are smaller than objects in earlier layers. For relations such as "AND" in object colors, the connections can capture whether for any node in the last diagram there is node of equal color in previous two diagrams. We improved explanation in section 'Multiplex Graph Network' of the revised version.
> >
> > 8. "What’s a cross-multiplexing gating function? If it’s a known concept, please provide a reference else explain":
> >
> > 'Cross-multiplexing gating function' is not a known concept but newly introduced in this paper. As discussed in the paper, this gating function accepts a set of summarised node embeddings as input, and output gating variables for each layer of node embeddings in the set. It is 'cross-multiplexing' because each embedding in the set is 'multiplexing' other embeddings in the set with gating variables that regulate which stream of information passes through. In the revised version, we added more explanations on this gating function.
> >
> >
> > Additional citations:
> > 1. Santoro, Adam, et al. "A simple neural network module for relational reasoning." Advances in neural information processing systems. 2017.
> > 2. Andreas, Jacob, et al. "Neural module networks." Proceedings of the IEEE Conference on Computer Vision and Pattern Recognition. 2016.
> > 3. Perez, Ethan, et al. "Film: Visual reasoning with a general conditioning layer." Thirty-Second AAAI Conference on Artificial Intelligence. 2018.
> > 4. Ren, Shaoqing, et al. "Faster r-cnn: Towards real-time object detection with region proposal networks." Advances in neural information processing systems. 2015.
> > 5. Eslami, SM Ali, et al. "Attend, infer, repeat: Fast scene understanding with generative models." Advances in Neural Information Processing Systems. 2016.

---

> ### Public Comment · ~Yuan_Yang2 · 2021-09-28
> **Question about confusing notation**
>
> In Section ''4 METHOD'', the first paragraph, line 4, it says ''for each diagram $d_i \subset C \cup A$'', but shouldn't diagrams be elements of $C$ and $A$?  If so, shouldn't it be $d_i \in C \cup A$ ?

---

### Official Review · AnonReviewer2 · 2019-10-31
**Official Blind Review #2**

**Rating:** 6

**Review:**

In this paper the authors solve for the task of Raven Progressive Matrices (RPM) reasoning. They do so by considering multiplexed graph networks. They present an architecture for the same. The basic premise is a combination of object level representation that is obtained by a method similar to region proposal and combining them with graph network. The approach uses gated graph networks that also uses an aggregation function. These are combined and result in node embeddings. Detailed analysis of the network is provided. This provides improved results over earlier WREN method. However, the performance is slightly lesser than another paper simultaneously submitted that achieves similar results. That approach uses transformer network for spatial attention while here the spatial attention is just based on object level representation.

Over all while the contribution is useful, not much analysis is provided on the interpretability of the results. For instance, the statistics in terms of the search space reduction as to how many subsets get pruned. Further, there may be subsets of graphs that could span across rows and columns. The decision in terms of restricting the reduction to span specific rows or columns may result in pertinent nodes also being pruned. Certain aspects that relate to object level representation are not very clear. I am not fully aware about results in this specific area and that may also be a reason for the same.

To conclude, I believe this paper provides a useful contribution by modeling the diagrammatic abstract reasoning as a graph based reasoning approach. The multiplex graph network could be a useful component that is also relevant for other problems. The paper provides sufficient analysis to convince us regarding the claims.

**Experience Assessment:**

I do not know much about this area.

**Review Assessment: Checking Correctness Of Derivations And Theory:**

I assessed the sensibility of the derivations and theory.

**Review Assessment: Checking Correctness Of Experiments:**

I assessed the sensibility of the experiments.

**Review Assessment: Thoroughness In Paper Reading:**

I read the paper at least twice and used my best judgement in assessing the paper.

---

> ### Author Response · Authors · 2019-11-11
> **Thank you for your valuable comments**
>
> Thank you for your valuable comments. In our revised version, we improved explanations of our models with more details. Here we address some of your concerns:
>
> 1. "the statistics in terms of the search space reduction as to how many subsets get pruned":
> We have added more statistics of the search space reduction experiments in Appendix D, such as the top-16 subsets with the highest gating values.
>
> 2. Further, there may be subsets of graphs that could span across rows and columns. The decision in terms of restricting the reduction to span specific rows or columns may result in pertinent nodes also being pruned:
>
> The subsets are not constrained to rows and columns. During search space reduction we only make the weak assumption that edges in the same subset must be adjacent (defined as two edges linking the same node). This allows for subsets other than rows and columns, such as diagonal of the matrix. The search space reduction experiments however give lower scores for subsets other than rows and columns. This is why we hard-gate only row and columns subsets in the final architecture. We explained this more clearly in the revised version.
>
> 3. "However, the performance is slightly lesser than another paper simultaneously submitted that achieves similar results. That approach uses transformer network for spatial attention while here the spatial attention is just based on object level representation":
>
> We have just noted this parallel submission and compared it with our results. We found that our model performs better for PGM dataset(89.6% against 88.2% in neutral split with beta=10). In their response to comments, they stated that their model achieved performance of 19.67% for RAVEN-10000, which is the public dataset we used in the experiments. We achieved 83.91% accuracy. They did not make it clear how did they obtain 50k samples for each figure configuration, but our guess is that they used the open-source code to generate more data than available in RAVEN-10000.

---

### Comment · AnonReviewer3 · 2019-10-23
**Another ICLR 2020 submission on PGM**

It's perhaps useful to compare this paper with Paper 1456, which also shows results on PGM and Raven.

This paper gets slightly better numbers.

A public comment there faults the paper for not showing results on generalization settings. I think this flaw is true for this paper as well.

---

> ### Author Response · Authors · 2019-10-23
> **Comparison with parallel submission and generalization experiments**
>
> Thank you very much for your comments. We were not aware of this paper but will definitely include it in revised version of our paper.
>
> We have in fact performed the generalization experiments, as discussed in section 5.4 and also in Appendix F. In section 5.4 we have performed experiments on generalization regime of 'interpolation' and 'extrapolation'. We put results of other generaliztion regimes in Appendix F because of the page limit. If there is anything unclear in the description, please let us know and we will improve it.
>
> Thanks again for taking time reading and commenting on our paper.

---

### Author Response · Authors · 2019-11-11
**Summary of changes made in the revised version**

In the revised version, we improved structuring and writing quality according to reviewers' comments. The major changes are:

1. Combined Figure 1 and 2 to give more space for other sections.
2. Moved parts of dataset description to Appendix to give more space for other sections.
3. Improved explanation of model naming in "Introduction" section.
4. Improved "Method" section. Specifically we improved explanations in 'Search Space Reduction', and added more explanation and motivation for multiplex edges and cross multiplexing gating function. We also added more details for the 'reasoning network'.
5. Added more results both in section 5.1 'search space reduction' and in Appendix D 'More details on search space reduction'.
6. Improved presentation in Appendix. Specifically we now represent architecture configurations with figures instead of tables, which makes it more reader friendly. We also added more detailed descriptions for all the modules.
7. Fixes typos and other minor issues such as formatting.

---

### Decision · Program_Chairs · 2019-12-19

**Decision:**

Accept (Poster)

**Comment:**

This paper a new method of constructing graph neural networks for the task of reasoning to answer IQ style diagrammatic reasoning, in particular including Raven Progressive Matrices. The model first learns an object representation for parts of the  image and then tries to combine them together to represent relations between different objects of the image. Using this model they achieve SOTA results (ignoring a parallely submitted paper) on the PGM and Raven datasets. The improvement in SOTA is subtantial.

Most of the critique made for the paper is on writing style and presentation. The authors seem to have fixed several of these concerns in the newly uploaded version of the paper. I will further request the authors to revise the paper for readability. However, since the paper presents both an interesting modeling and improved empirical results, I recommend acceptance.